# Differentiation between Impacted and Unimpacted Microbial Communities of a Nitrogen Contaminated Aquifer

Justin G. Morrissy [1,*], Suzie M. Reichman [2,3], Matthew J. Currell [1,4], Aravind Surapaneni [5,6], Mallavarapu Megharaj [7,8], Nicholas D. Crosbie [9], Daniel Hirth [10], Simon Aquilina [11], William Rajendram [12] and Andrew S. Ball [6]

1. School of Engineering, RMIT University, GPO Box 2476, Melbourne, VIC 3001, Australia
2. Centre for Anthropogenic Pollution Impact and Management (CAPIM), University of Melbourne, Parkville, VIC 3010, Australia
3. School of BioSciences, University of Melbourne, Parkville, VIC 3010, Australia
4. Water: Effective Technologies & Tools Research Centre, RMIT University, GPO Box 2476, Melbourne, VIC 3001, Australia
5. South East Water, Frankston, VIC 3199, Australia
6. ARC Training Centre for the Transformation of Australia's Biosolids Resource, School of Science, RMIT University, Bundoora West, VIC 3083, Australia
7. Global Centre for Environmental Remediation (GCER), Faculty of Science, ATC Building, The University of Newcastle, Callaghan, NSW 2308, Australia
8. Cooperative Research Centre for Contamination Assessment and Remediation of Environment (CRC CARE), The University of Newcastle, ATC Building, Callaghan, NSW 2308, Australia
9. Melbourne Water, Docklands, VIC 3008, Australia
10. BlueSphere Environmental, Southbank, VIC 3006, Australia
11. Gippsland Water, Traralgon, VIC 3844, Australia
12. Greater Western Water, Sunbury, VIC 3429, Australia
* Correspondence: s3705476@student.rmit.edu.au

**Abstract:** Nitrogen contamination is ubiquitous across the globe; as a result of this, the need to understand and predict the extent and effects of nitrogen contamination on microbial ecosystems is increasingly important. This paper utilises a dataset that provides a rare opportunity to observe varying contamination conditions in a single aquifer and understand the differences between potential background bores and two different types of contamination spread across the other bores. Using physicochemical and microbiological community analysis, this paper aims to determine the impacts of the two contaminants, nitrate and ammonia, on the microbial communities and the differences between polluted and physicochemical background bores. Total nitrogen (N) varied by a factor of over 2000 between bores, ranging from 0.07 to 155 mg $L^{-1}$. Nitrate ($NO_3^-$) concentrations ranged from 150 to <0.01 mg $L^{-1}$; ammonium ($NH_4^+$) concentrations ranged from 26 to <0.1 mg $L^{-1}$. MANOVA analysis confirmed an overall significant relationship ($p = 0.0052$) between N variables and the physicochemical data (or status) of the three areas of contamination dubbed 'contamination zones'. The contamination zones were defined by no known presence of contamination in the uncontaminated bores, the presence of $NO_3^-$ contamination and the presence of $NO_3^-$ and $NH_4^+$ contamination. PERMANOVA analysis confirmed that there was an overall significant difference in the microbial communities between the three contamination zones ($p = 0.0002$); however, the presence of $NH_4^+$ had a significant effect ($p = 0.0012$). In general, the nitrate-contaminated bores showed a decrease in the abundance of individual OTUs. We further confirmed that $NH_4^+$ contamination had a significant relationship with an increased percentage of abundance occupied by the Planctomycetota phylum (specifically the *Candidatus Brocadia* genus). It was found that one of the two background bores (BS-004) was likely also representative of natural microbial background, and another (BS-002) showed characteristics that may be representative of past or intermittent contamination. This paper demonstrates a possible way to determine the microbial background and discusses the potential uses for this information.

**Keywords:** nitrogen contamination; groundwater; biogeochemistry; microbial biochemistry; microbial ecology; denitrification; biological N decomposition; ANAMMOX; groundwater microbial community characterisation

## 1. Introduction

Pollution of land and water is a trillion-dollar issue globally, contributing to the loss of biodiversity, acidification of oceans, eutrophication of water sources and global climate change [1,2]. Many contaminants have been so widely present, for such a long period, that it has become challenging to determine what ecosystems and environments were before their presence. Nitrogen contamination is currently less prominent in media coverage and public debate than global climate change or plastic pollution; however, it is one of the oldest and most widespread environmental contaminants. The impact of nitrogen pollution includes an increasing incidence of eutrophication globally, as well as an increased incidence of respiratory distress and methemoglobinaemia in humans [3,4]. Nitrate is also one of the main sources of groundwater contamination [5]. Nitrogen contamination originates from many sources: overuse of fertilisers in agriculture, leakage of privately owned wastewater tanks, leakage of wastewater treatment plant (WWTP) infrastructure and leakage of chemical by-products from industrial applications, to name a few [6]. The artificial fixation of nitrogen has more than doubled natural global nitrogen fixation from ~203 $Tg \cdot N \cdot yr^{-1}$ to ~413 $Tg \cdot N \cdot yr^{-1}$; however, the rate of conversion of reactive nitrogen back into atmospheric nitrogen has been unable to keep up at ~339 $Tg \cdot N \cdot yr^{-1}$ [6]. As a result, nitrogen is accumulating in the environment, which can be seen in the increasing rates of eutrophication and multiple papers discussing the different effects of nitrogen pollution [7–9].

### 1.1. Physicochemical Background

Establishing background conditions is essential to understanding the extent and/or effect of any contamination in a region. Without a background assessment, it becomes difficult to differentiate between areas with naturally different physicochemical or microbial characteristics and areas that have been altered due to anthropogenic influence. Additionally, situations where remediation efforts are put in place in areas where they are not necessarily needed can potentially be avoided by first establishing background conditions. Determining the background of physicochemical properties can be exceptionally difficult in situations where contamination is ubiquitous. As a result of this, other factors need to be considered to assist in the differentiation between background and contaminated conditions.

### 1.2. Microbial Background

Microbial communities represent a vast reservoir of potential, in their capability to perform natural ecosystem functions, such as their huge influence on all the major and minor global nutrient cycles, their ability to adapt to changes in their ecosystems, and their ability to resist and degrade various pollutants [10,11]. The adaptability of microbial communities is often viewed as one of their most remarkable traits, especially in the fields of contamination and remediation. However, the consequences that come about through changes to ecosystem structure and function because of adaptation are rarely considered. This is exacerbated by a lack of ability to determine the differences between unimpacted and impacted microbial communities.

Physicochemical testing and background are well-established benchmarks, have been thoroughly explored within the literature by authors such as Reimann and Garrett [12] and Panno et al. [13] and are well-understood methods for assessing the impacts of contamination in the environment. In contrast, our understanding of microbial ecosystems and their background states, in groundwater or otherwise, is lacking. Microbial background

is also far more complex than that of a physicochemical background; for example, in the data set presented in this paper, there are over 6000 operational taxonomic units (OTUs), all of which may react differently under various conditions. Delving straight into the complexity of understanding the natural ranges of 6000 microbial populations and how they interact with each other, and the constantly changing physicochemical and environmental conditions, even in environments considered stable, such as groundwater, is a huge undertaking. Initially, to mitigate some of this complexity, looking at some of the larger elements of the microbial communities is likely a good starting point. Additionally, it is likely that the exact structure of the microbial ecosystem is constantly slightly shifting; however, the major ecosystem functions should remain relatively consistent. Steube et al. [14] and Griebler et al. [15] discuss the importance of determining the background for both physicochemical and ecological parameters for groundwater and speculate about general parameters, such as biomass and community structure, that could be used to calculate the background for microbial communities. However, despite both papers using statistical analysis to establish background thresholds for the physicochemical properties, neither was able to make progress towards definitive methods for differentiating between contaminated and background microbial communities. The data collected for this paper presents an opportunity to study both unimpacted communities and communities impacted by several pollution sources simultaneously across a relatively small region within a single aquifer. The changes in the microbial communities across different contaminated and uncontaminated regions allow us to determine the characteristics of the different impacted and unimpacted microbial communities and use this to differentiate between them.

The aims of this paper are to:

1. Determine how nitrogen contamination influences the structure and ecosystem functions of the microbial communities of a nitrogen-contaminated aquifer.
2. Use microbial community data and physicochemical data to explore how to differentiate between impacted and unimpacted microbial communities.
3. Demonstrate the value of microbial functional group analysis in the characterisation of the impact and extent of groundwater contamination.

Our previous work in Morrissy et al. [16] identified that nitrogen pollution had an effect on the microbial communities at the Boneo site in Victoria, Australia; additionally, multiple papers have discussed the structure of microbial communities and how they change over different conditions [17–19]. The importance of this work lies in the novelty of utilising a relatively small dataset to show the effects of pollution on the major ecosystem functions of microbial communities in groundwater and how we can use this information to predict contaminant impact, fate, and behaviour in groundwater. This should be of particular interest to industries involved in contamination as it shows how much information can be gained from utilising 16S sequencing even for a relatively short-term and inexpensive sampling campaign.

## 2. Materials and Methods

Other than the data analysis, the methods adopted here are the same as those outlined in Morrissy et al. [16] (with some minor rewording), as this current paper is a further analysis of the data (towards the objectives outlined above) as a part of a larger research project.

### 2.1. Study Area

A detailed description of the sampling site (Figure 1) can be found in Adebowale et al. [20] and McCance et al. [21]. Briefly, the site is approximately 80 km southeast of Melbourne, Australia. The site represents a typical example of a region with multiple current and historical nitrogen contamination sources. The main nitrogen groundwater plume was centred around the wastewater treatment plant (WWTP); however, due to the up-gradient intensive agriculture (market garden farms) and down-gradient cattle grazing paddocks, the normal background concentration of nitrogen and thus the extent of influence of the WWTP versus other sources was challenging to determine. Delineation of

current and historical contamination plumes at the site are discussed in McCance et al. [21] and McCance et al. [22]; the main plume extends from bore site RB11 across RB12 and through RB13/14 and past RB17/18 (Figure 1).

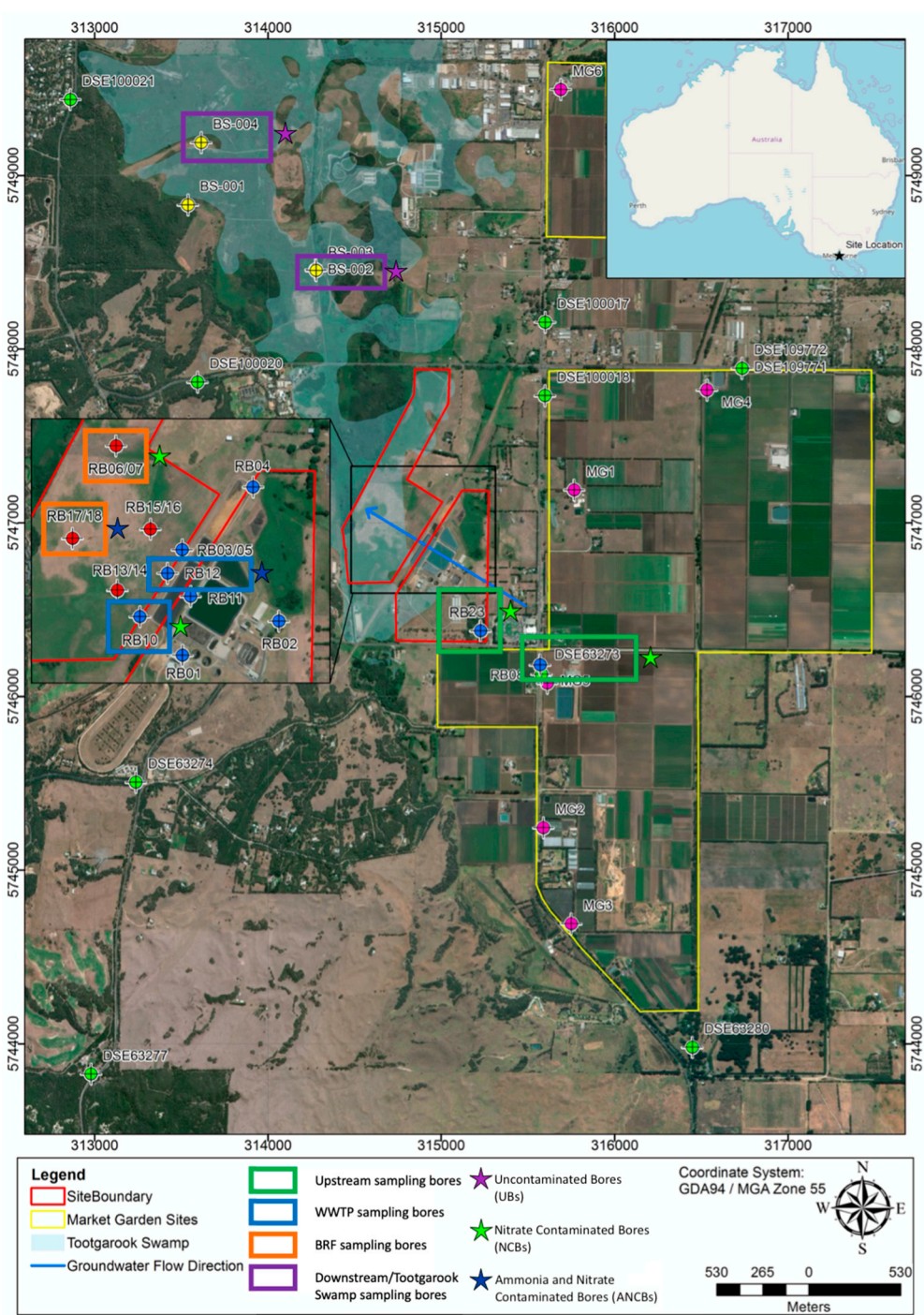

**Figure 1.** Site location and layout map. BRF = Browns Road Farm.

The Quaternary Bridgewater Formation was the primary aquifer of concern at this site. It comprises carbonate-cemented aeolian sands and is unconfined and up to 100 m thick [23]. Across the region, the hydraulic conductivity of the aquifer averages approximately 20 m/day [23]. Further details about the aquifer setting and site history can be found in McCance et al. [21] and McCance et al. [22].

## 2.2. Sampling Timeline and Locations

For the duration of the study, consisting of three sampling campaigns, 10 bores were sampled and a total of 24 groundwater samples were taken. All bores were not able to be sampled on every sampling campaign (hence 24 samples instead of 30); all samples were taken in triplicate (resulting in a total of 72 subsamples). The three sampling campaigns took place in August 2018, November 2018 and May 2019 (more detail on the bores sampled in each campaign can be found in Morrissy et al. [16] Table 1). The locations of the sampled bores are shown in Figure 1. Further details about the bores are described in Adebowale et al. [20]. The groundwater flowed from southeast to northwest. Bores DSE63273 and RB23 are both up-gradient from the WWTP (in terms of the regional groundwater flow direction—see McCance et al. [21]). Bores RB10 and RB12 are adjacent to each other within the WWTP boundary; RB17/18 and RB06/07 are nested bores (one shallow, one deeper) within the Browns Road Farm down-gradient of the WWTP. Bores BS02 and BS04 are located further north along the groundwater flow direction, within the Tootgarook Swamp, within a conservation reserve. These sites currently show no indication of nitrate or any other contamination from either the WWTP or agriculture. Further details about the groundwater flow paths can be found in McCance et al. [22].

**Table 1.** The physiochemical ranges and values of the ten groundwater wells sampled for this project. Units in mg $L^{-1}$ unless otherwise stated.

| Min-max Physiochemical Properties | DSE63273 | RB23 | RB10 | RB12 | RB17 | RB18 | RB06 | RB07 | BS-002 | BS-004 |
|---|---|---|---|---|---|---|---|---|---|---|
| Dissolved Oxygen | 4.47 | 4.77–5.92 | 0.22–0.51 | 0.15–0.47 | 0.2–0.64 | 0.17–0.56 | 0.12–0.61 | 0.22–0.76 | 0.24–0.74 | 0.17–0.69 |
| EC (field) (uS/cm) | 2945 | 1434–1487 | 2357–2659 | 2192–2641 | 1486–1723 | 1309–1480 | 1915–2633 | 3342–3503 | 940–1004 | 718–776 |
| pH (Field) (unitless) | 6.70 | 6.93–7.27 | 6.68–6.73 | 6.69–6.81 | 6.84–7.01 | 6.87–7.24 | 6.79–7.19 | 6.73–6.78 | 7.37–7.43 | 7.44–7.77 |
| Redox Potential (Field) (mV) | 107.0 | 28.9–111.3 | 21.1–61.2 | 23.3–68.7 | 29.1–77.2 | 20.7–79.9 | 33.4–35.9 | −8.4–24.0 | −59.6–5.5 | −106.9–−83.5 |
| Alkalinity (total) as CaCO$_3$ | 315 | 170–270 | 430–610 | 450–560 | 350–370 | 350–370 | 270–330 | 410–420 | 240–250 | 210–210 |
| Bicarbonate, as Bicarbonate | 385 | 210–329.4 | 530–744.2 | 550–683.2 | 430–450 | 430–450 | 330–400 | 490–510 | 300–300 | 250–260 |
| Sulfate as SO$_4$$^{2-}$ | 410 | 24–45 | 580–760 | 330–530 | 220–330 | 130–190 | 490–1100 | 1500–1500 | 7–8 | 26–34 |
| Anionic Strength (meq/L) | 32 | 13–17 | 29–34 | 26–29 | 17–20 | 14–16 | 22–37 | 48–48 | 9–10 | 7–8 |
| Cationic Strength (meq/L) | 28.5 | 12–14 | 27–34 | 24–31 | 18–18 | 13–15 | 18–37 | 36–36 | 8–9 | 7–7 |
| Calcium | 390 | 150–170 | 350–430 | 230–320 | 200–200 | 130–170 | 210–480 | 470–570 | 83–84 | 47–51 |
| Magnesium | 41 | 15–19 | 50–69 | 48–61 | 27–32 | 22–25 | 34–63 | 70–81 | 11–11 | 25–28 |
| Sodium | 135 | 75–85 | 110–140 | 120–160 | 120–130 | 100–110 | 110–170 | 160–210 | 78–82 | 49–52 |
| Potassium | 1.2 | 1–1 | 11–14 | 31–40 | 0.8–0.8 | 3–3.5 | 0.8–0.8 | 1–7 | 1–1.3 | 4–4 |
| Chloride | 230 | 150–170 | 170–210 | 170–210 | 150–160 | 110–140 | 180–250 | 300–320 | 160–170 | 93–100 |
| Bromide | 1600 | 680–970 | 3700–4700 | 1200–1400 | 530–830 | 510–600 | 700–940 | 1500–1600 | 380–450 | 230–280 |
| Ammonia as N | 0.15 | <0.1–<0.1 | <0.1–<0.1 | 5.5–26 | <0.1–<0.1 | 3.3–4.6 | <0.1–<0.1 | 0.1–0.1 | 0.1–0.1 | 0.2–0.2 |
| Nitrate (as N) | 150 | 64–83 | 14–54 | 40–63 | 12–15 | 11–18 | 7.5–18 | 0.22–0.22 | 0.01–0.01 | <0.01–<0.01 |
| Nitrogen (Total Oxidised) | 155 | 64–88 | 15–54 | 40–63 | 12–15 | 11–15 | 7.15–18 | 0.26–0.26 | 0.01–0.01 | <0.01–<0.01 |
| Nitrogen (Total) | 155 | 65–88 | 15–55 | 60–76 | 13–16 | 18–20 | 8.2–20 | 0.9–1.2 | 0.07–0.2 | 0.3–0.66 |
| TOC | 3.7 | 1.3–2.5 | 10–13 | 10–12 | 1.6–7.1 | 1.3–6.1 | 4.4–8.1 | 13–16 | 2.6–2.6 | 2.7–3.1 |
| Organic Nitrogen, as N | 0.6 | 0.2–0.4 | 0.8–1.2 | 0.2–0.2 | 0.4–0.6 | 0.1–0.1 | 0.7–0.8 | 0.7–1.1 | 0.1–0.2 | 0.1–0.4 |
| Kjeldahl Nitrogen Total | 3.2 | 0.2–0.7 | 0.8–1.3 | 5.5–26 | 0.5–2 | 3.1–8.3 | 0.8–2.2 | 0.9–0.94 | 0.06–0.2 | 0.3–0.46 |
| Phosphorus | 0.009 | <0.005–<0.005 | <0.005–<0.005 | 0.05–0.05 | <0.005–<0.005 | 0.006–0.006 | <0.005–<0.005 | 0.08–0.08 | <0.005–<0.005 | 0.05–0.05 |
| TDS | 2250 | 960–1200 | 1700–1900 | 1400–1600 | 1000–1200 | 790–840 | 1200–2100 | 2700–2800 | 440–480 | 300–360 |
| Iron (Filtered) | 0.02 | 0.003–0.003 | 0.015–0.015 | 0.009–0.009 | 0.003–0.003 | 0.003–0.003 | <0.1–<0.1 | 0.9–1.2 | 0.096–0.13 | 0.57–1.4 |
| Molybdenum (Filtered) | <0.001 | 0.0006–0.0006 | <0.001–<0.001 | <0.001–<0.001 | 0.0002–0.0002 | 0.0001–0.0001 | <0.001–<0.001 | <0.001–<0.001 | 0.0002–0.0002 | 0.004–0.0055 |
| Nickel (Filtered) | <0.001 | 0.0011–0.0011 | 0.004–0.011 | 0.004–0.0054 | 0.00078–0.003 | 0.00059–0.00059 | 0.002–0.002 | 0.001–0.002 | 0.002–0.0051 | 0.00018–0.00018 |
| Zinc (Filtered) | 0.0295 | 0.006–0.014 | 0.01–0.013 | 0.0086–0.015 | 0.0043–0.011 | 0.004–0.007 | 0.005–0.032 | 0.01–0.01 | 0.0026–0.008 | 0.0007–0.003 |
| Bore Depth (CS) (m) | 25.1 | 11.1 | 5.3 | 5.1 | 4.7 | 10.8 | 9.1 | 4.6 | 26.4 | 5.5 |
| Number of samples | 1 | 3 | 3 | 3 | 3 | 3 | 2 | 2 | 2 | 2 |

## 2.3. Sampling Techniques, Technology and Guidelines

Groundwater samples were collected in accordance with environmental protection agency (EPA) Victoria guidelines [24,25] using low flow sampling techniques. For monitoring bores, the standing water level was first measured using a Solinst interface probe, and then continuously monitored during low-flow pumping, along with field physio-chemical parameters (electrical conductivity, oxidation-reduction potential, dissolved oxygen, tem-

perature and pH). These parameters were monitored using a multi-parameter field probe (YSI Pro Plus or HACH HQ40D) [20]. Samples were collected following stabilisation of these parameters.

### 2.4. Major Ions and Nutrients Analysis

Samples for analysis of major ions and nutrients were collected into bottles provided by Australian Laboratory Services (ALS laboratory) and delivered to the laboratory on the same day for analysis, using standard analytical techniques required under the National Association of Testing Authorities (NATA) accreditation. Sample duplicates, triplicates and field blanks were collected according to the standard operating procedures (Table S1) and analysed for quality assurance; all reported data met the necessary reporting thresholds of these methods.

### 2.5. Microbial Analysis

#### 2.5.1. Sampling

Samples for microbial analysis were collected in sterilised 1 L, round, borosilicate amber glass laboratory bottles. Bottles were sterilised in an autoclave at standard temperature and pressure before being sealed and transported to the sampling location. After collection, samples were transported back to the lab and filtered using a Microfil, mixed cellulose esters, 0.22 μm white gridded, sterile filter within a sterile filter apparatus. Filtered samples were stored at $-20\ °C$ in a freezer within 24 h of collection.

#### 2.5.2. DNA Extraction and Sequencing

Once the samples from all sampling rounds were processed, they were removed from the freezer, and DNeasy PowerWater Kits used to extract the DNA from the filtered samples. After extraction, samples were processed for 16S rRNA sequencing using the Nextera $^®$ XT Index Kit (Illumina, San Diego, CA, USA) as outlined in the 16S Metagenomic Sequencing Library Preparation guide provided by Illumina. The DNA from the library was quantified using Qubits 2.0 Fluorometer (Life Technologies, Carlsbad, CA, USA) and 2100 Bioanalyzer (Agilent Technologies, Santa Clara, CA, USA). The samples were pooled and run in a MiSeq platform (Illumina, San Diego, CA, USA) at the School of Science, RMIT University [26].

#### 2.5.3. File Preparation

Following sequencing, the raw files were copied from the MiSeq machine and loaded into R studio along with all the collected environmental data (including the major ions and nutrients), where further processing and analysis were completed. Raw files were trimmed, chimeric reads were removed, and reads were grouped into operational taxonomic units (OTUs) and assigned taxonomic classifications. This was performed using the Applecorn script (https://github.com/MonashBioinformaticsPlatform/applecorn) (accessed on 13 September 2019), and taxonomic classifications used the silva (132) database for reference.

#### 2.5.4. Data Analysis

Multiple MANOVAs were used to determine significant differences between the contamination zones and multiple PERMANOVAs were used to determine the effects of the contamination zones and nitrogen molecules on the microbial communities. Sunburst charts were made in excel using the OTU abundance data that was exported from R studio (a list of the primary packages used throughout R studio analysis can be seen in Table S2). The term OTU will be used in reference to the lowest taxonomic level of analysis when referring to our data throughout the rest of this paper. The method of grouping OTUs is typically accurate up until the genus level; however, it can underestimate species richness (i.e., each OTU may represent more than one species) [27]. OTUs are preferable in this case as they are more inclined to slightly overestimate the impact of a contaminant rather than underestimate the impact. Additionally, there is no real effect on the comparisons

between samples as they all use the same method, the effect would only come into play when comparing richness data to other datasets.

## 3. Results

### 3.1. Physicochemical Properties

Total N varied by a factor of over 2000, ranging from 155 to 0.07 mg $L^{-1}$, with the highest concentration found in bore DSE63273 (Table 1). The high concentration of total N at this site was likely due to being immediately downstream from an agricultural region and adjacent to a poultry manure stockpile [20]. The lowest concentration of total nitrogen was found in the BS-002 bore in the Tootgarook Swamp; this bore was chosen as a potential representative of uncontaminated groundwater (with respect to N) in the region.

Nitrate ($NO_3^-$) concentrations ranged from 150 to <0.01 mg $L^{-1}$ across the sampled bores, with the highest concentration also found in bore DSE63273. The lowest concentration of $NO_3^-$, which was below the detection limit, was found in bore BS-004, in the Tootgarook Swamp, which is further evidence of the minimal anthropogenic impact at this site. Together with bore BS-002, the two Tootgarook Swamp bores, from the site history and physicochemical evidence, were considered to represent (to the best extent possible) unimpacted sites. This considers the low nitrogen concentrations and their location within the aquifer and region. These concentrations also coincide with concentrations seen in historical data of the now-contaminated bores from before they were contaminated; it should be noted that both the agricultural region and the WWTP predate the beginning of the historical data (unpublished data). Ammonium ($NH_4^+$) concentrations ranged from 26 to <0.1 mg $L^{-1}$ across the bores, with the highest $NH_4^+$ concentration found in RB12. The concentration of $NH_4^+$ was below detection limits in bores RB23, RB10, RB17 and RB06 and at or close to the detection limits in bores BS-002 and BS-004. The highest concentrations of $NH_4^+$ was within and immediately downstream from the RB12 side of the WWTP.

Nitrate ($NO_3^-$) concentrations ranged from 150 to <0.01 mg $L^{-1}$ across the bores, with the highest $NO_3^-$ concentration found in DSE63273. The concentration of $NO_3^-$ was at or close to the detection limits in bores BS-002 and BS-004. The highest concentrations of $NO_3^-$ were immediately downstream from the agricultural region with the concentrations generally decreasing the further downstream they were, with the exception of a spike at the location of the RB12 bore. Total nitrogen shows similar trends to the $NO_3^-$; however, no samples were at or below the detection limits. Table 1 also shows that most of the nutrients in bores BS-002 and BS-004 had lower concentrations than the other bores, while RB07 and BS-004 had somewhat higher concentrations of iron (Fe) than the other bores.

### 3.2. Nitrogen and Physicochemical Properties in Contamination Zones

The analysis of the data presented in Morrissy et al. [16] allowed us to identify three 'contamination zones' which were defined by no known presence of contamination in the uncontaminated bores (UB's) (BS-002 and BS-004), the presence of $NO_3^-$ contamination (NCBs) (bores DSE63273, RB23, RB10 and RB6/7) and the presence of $NO_3^-$ and $NH_4^+$ contamination (ANCBs) (bores RB12 and RB17/18). Using MANOVA analysis of the physicochemical data with the N variables as the dependent variables, we determined if there was any statistically significant difference between the contamination zones with regard to N contamination (Table 2). The MANOVA shows an overall significant relationship ($p = 0.0052$) between N variables and contamination zones. Pairwise analysis shows a significant relationship between the NCBs and the UBs ($p = 0.0079$) and ANCBs ($p = 0.0133$), and a significant relationship between the ANCBs and the UBs ($p = 0.0438$).

**Table 2.** P and Pr (>F) values from MANOVA and PERMANOVA statistical tests determining the differences between all data and pairwise analysis of the 3 contamination zones.

| | MANOVA (p) | PERMANOVA (Pr(>F)) | |
|---|---|---|---|
| **Analysis Groups** | **All Nitrogen Variables** | **Contamination Zones** | **Ammonia ($NH_4^+$)** |
| all | 0.0052 | 0.0002 | 0.0012 |
| UB-ANCB | 0.0438 | 0.0099 | 0.0453 |
| UB-NCB | 0.0079 | 0.4048 | 0.4323 |
| NCB-ANCB | 0.0133 | 0.0001 | 0.0006 |

### 3.3. Nitrogen and Microbial Communities in Contamination Zones

When analysing the microbial community data, we used PERMANOVA analysis to show that there was an overall significant difference between the three contamination zones (Pr (>F) = 0.0002) (Table 2). We also tested to see if the presence of $NH_4^+$ had a significant effect (Pr (>F) = 0.0012) (Table 2). This showed that although much of the variance can be explained by the presence of $NH_4^+$, more of the variance can be explained by the contamination zones. The pairwise analysis of the contamination zones shows that there are significant differences between the ANCBs and both the UBs (Pr (>F) = 0.0099) and NCBs (Pr (>F) = 0.0001), but not between the UBs and the NBCs (Pr (>F) = 0.4048). The fact that the contamination zones explained more of the variance indicates that the effect of the plume extending from the WWTP may not be solely attributed to $NH_4^+$. Pairwise analysis also confirms the original test results that most, but not all, of the variance between the contamination zones can be explained by the presence of $NH_4^+$ (Pr (>F) UB-NCB = 0.4323, Pr (>F) UB-ANCB = 0.0453 and Pr (>F) NCB-ANCB = 0.0006).

### 3.4. Major Ecosystem Functions

Determining the ecosystem functions of the ecosystems in different contamination zones and bores requires integrating both the microbial community data and the physico-chemical data to understand the functions the major OTUs perform and their purpose. To gain a snapshot of the functions performed by the most abundant microorganisms in the different bores for the study area, the following will discuss the 20 OTUs with the highest abundance, up to 50% (Table 3).

**Table 3.** The richness, diversity and evenness estimators and the percentage commonality of OTUs between bores.

| Min Max Microbial Community Descriptors | DSE63273 | RB23 | RB10 | RB12 | RB17 | RB18 | RB06 | RB07 | BS-002 | BS-004 |
|---|---|---|---|---|---|---|---|---|---|---|
| Non-Chimeric Sequences | 6403 | 12,150–26,674 | 1584–21,782 | 7353–23,760 | 1542–21,920 | 1691–30,838 | 1964–2284 | 15,628–36,019 | 9588–33,592 | 1809–13,032 |
| OTUs | 1289 | 1153–1263 | 263–2590 | 1705–2391 | 415–1643 | 398–1591 | 556–758 | 2894–2906 | 940–1534 | 379–1882 |
| Total OTUs | 1289 | 1725 | 2926 | 3055 | 2276 | 2051 | 1035 | 3711 | 1682 | 2146 |
| ACE | 1429 | 1227–1309 | 291–2717 | 1907–2551 | 432–1790 | 409–1703 | 578–837 | 3075–3099 | 994–1589 | 389–2066 |
| Chao1 estimated no species | 1524 | 1260–1404 | 779–3333 | 2069–5870 | 574–1772 | 593–1916 | 858–1664 | 3345–6421 | 1039–1614 | 610–2090 |
| Gini-Simpson | 0.01 | 0.01–0.04 | 0.01–0.11 | 0.04–0.13 | 0.02–0.1 | 0.02–0.1 | 0.01–0.01 | 0.01–0.01 | 0.03–0.05 | 0.01–0.02 |
| Shannon rare | 5.41 | 4.6–5.72 | 3.56–6.58 | 3.58–5.35 | 3.98–4.75 | 3.92–4.91 | 5.19–5.55 | 6.34–6.45 | 4.65–4.92 | 4.87–6.03 |
| Pielou's Evenness | 0.76 | 0.64–0.81 | 0.64–0.84 | 0.48–0.7 | 0.54–0.79 | 0.54–0.82 | 0.82–0.84 | 0.8–0.81 | 0.63–0.72 | 0.8–0.82 |
| **Percent of OTUs in Common** | **DSE63273** | **RB23** | **RB10** | **RB12** | **RB17** | **RB18** | **RB06** | **RB07** | **BS-002** | **BS-004** |
| DSE63273 | | 12.21 | 16.47 | 16.77 | 14.08 | 13.72 | 12.22 | 23.12 | 17.52 | 22.46 |
| RB23 | 12.21 | | 24.26 | 22.03 | 27.54 | 34.14 | 26.14 | 21.69 | 33.82 | 20.25 |
| RB10 | 16.47 | 24.26 | | 52.62 | 40.48 | 32.30 | 20.65 | 41.54 | 17.16 | 23.83 |
| RB12 | 16.77 | 22.03 | 52.62 | | 42.24 | 36.63 | 19.24 | 42.95 | 16.16 | 23.98 |
| RB17 | 14.08 | 27.54 | 40.48 | 42.24 | | 49.83 | 24.05 | 31.35 | 18.29 | 21.72 |
| RB18 | 13.72 | 34.14 | 32.30 | 36.63 | 49.83 | | 26.11 | 27.37 | 25.73 | 21.16 |
| RB06 | 12.22 | 26.14 | 20.65 | 19.24 | 24.05 | 26.11 | | 18.03 | 20.81 | 17.25 |
| RB07 | 23.12 | 21.69 | 41.54 | 42.95 | 31.35 | 27.37 | 18.03 | | 21.60 | 38.96 |
| BS-002 | 17.52 | 33.82 | 17.16 | 16.16 | 18.29 | 25.73 | 20.81 | 21.60 | | 23.64 |
| BS-004 | 22.46 | 20.25 | 23.83 | 23.98 | 21.72 | 21.16 | 17.25 | 38.96 | 23.64 | |

### 3.4.1. Nitrogen Contaminated Bores (NCB)

*Bore DSE63273*

The 20 OTUs with the highest abundance in the DSE63273 bore made up 41.4% of the total abundance in the bore; 15 of the top 20 were Proteobacteria (30.5%) (Figure 2). Within the Proteobacteria there were two sulphur (S)-reducing Deltaproteobacteria (Family *Desulfobulbaceae*). The remaining proteobacteria were Gammaproteobacteria consisting of eight methanotrophs (16.9%) (Family *Methylomonaceae*), two chemotrophs with known resistance to metals (Genus *Cupriavidus*), two known to mineralise aromatic compounds (Genus

*Acinetobacter*) and one which is associated with agricultural activity, N fixation, S oxidation and can be pathogenic (Genus *Stenotrophomonas*) [28–30]. Additionally present were a class of the Chloroflexota phylum known to be chemoheterotrophic (Class *Anaerolineae*) [31], an order of the cyanobacteria phylum capable of $NO_3^-$ reduction (Order *Melainabacteria*) [32], two Rokubacteria (NC10) thought to couple anaerobic methane ($CH_4$) oxidation with nitrite ($NO_2^-$) reduction (Family *Methylomirabilaceae*) [32] and a completely uncharacterised microorganism. The DSE63273 bore resides immediately downstream from an agricultural region and beside a road. Thus, the most abundant microorganisms in bore DSE63273 appeared to be focused around methanotrophs, S reduction and N respiration.

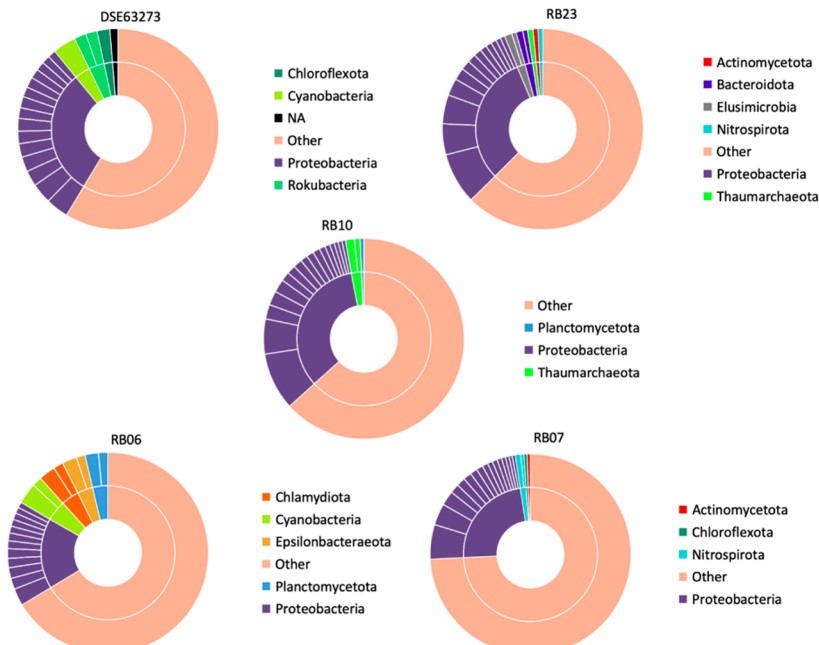

**Figure 2.** Sunburst chart of 20 microorganisms (OTUs) with the highest abundance in the nitrate-contaminated bores (NCBs) (DSE63273, RB23, RB10, RB06, RB07) separated by bore and organised by phylum. 'Other' represents the proportion of OTUs not represented by the 20 most abundant OTUs. The inner ring of the sunburst chart represents the phylum taxonomic level and the inner ring represents the individual OTUs.

*Bore RB23*

The 20 OTUs with the highest abundance in the RB23 bore made up 37.4% of the total abundance in the bore; 13 of the top 20 were Proteobacteria (31.2%) (Figure 2). Within the Proteobacteria there were two OTUs of *Desulfarculaceae*, three Gammaproteobacteria with known denitrifiers in the family (Family *Burkholderiaceae*) [33], and eight Alphaproteobacteria consisting of three complete denitrifiers (Genus *Rhodoplanes*) [34], two S and Fe reducers (7.4%) (Genus *Rhodobacter*) [35,36] and three possible degraders of aromatic compounds (14.3%) (Family *Sphingomonadaceae*). In addition, there was one Fe oxidiser (Class *Acidimicrobiia*) [37], four fermentative bacteria from the Bacteroidota and Elusimicrobia phyla (Family *Chitinophagaceae* and Order *Elusimicrobia*) [38,39], one $NO_2^-$ oxidiser in the Nitrospirota phylum (Genus *Nitrospira*) [40] and one $NH_4^+$ oxidising archaea in the Thaumarchaeota phylum (Family *Nitrosopumilaceae*) [41]. Bore RB23 was situated in a small grove of trees slightly upstream from the WWTP and further downstream than bore DSE63723 from the agricultural region. The most abundant microorganisms in bore RB23 focus on $NO_3^-$ reducers, denitrifiers, $NH_4^+$ oxidisers and an equal emphasis on fermentation, Fe and S reduction, and degradation of aromatic compounds.

*Bore RB10*

The 20 OTUs with the highest abundance in the RB10 bore made up 36.7% of the total abundance in the bore; 17 of the top 20 were Proteobacteria (33.8%) (Figure 2). Within the Proteobacteria there were 12 Gammaproteobacteria; three OTUs of *Burkholderiaceae*, one OTUs of *Nitrosomonadaceae* and eight opportunistic pathogens from the *Pseudomonas* (5.5%) and *Legionella* (18.4%) genera; *Pseudomonas* is also capable of $NO_3^-$ reduction [42,43]. Alphaproteobacteria has two OTUs of *Rhodoplanes* [34], two S reducers (Genus *Myxococcales*) [44], and a N fixer (Family *Rhizobiaceae*) [45]. Additionally, there were two OTUs of *Nitrosopumilaceae* and one ANAMMOX bacteria (Family *Brocadiaceae*). Bore RB10 resides within the bounds of the WWTP and immediately adjacent to a sludge drying pan. The most abundant microorganisms in bore RB10 were known opportunistic pathogens in humans and microorganisms involved with the N cycle.

*Bore RB06*

The 20 OTUs with the highest abundance in the RB06 bore made up 33.7% of the total abundance in the bore; 14 of the top 20 were Proteobacteria (17.0%) (Figure 2). The Proteobacteria phylum was represented by four OTUs of *Sphingomonadaceae*, two OTUs of *Enterobacteriaceae*, two OTUs of *Immundisolibacter* and a single OTU of *Pseudomonas*, *Stenotrophomonas*, an OTU capable of S reduction and acetate, ethanol and propanol degradation (Genus *Desulfuromonas*) [46] and a sulphate ($SO_4^{2-}$) oxidiser (Genus *Thermithiobacillus*) [47]. Additionally, there were two OTUs of *Sericytochromatia*, two OTUs of Arcobacter, two OTUs of *Candidatus Brocadia* and two unknown OTUs from the *Chlamydiota* phylum. Bore RB06 was situated in the Brown's Road Farm slightly further downstream from the WWTP than RB17/18 and further north. The most abundant microorganisms in bore RB06 were aromatic compound degraders, S and N-related OTUs and a few pathogenic OTUs.

*Bore RB07*

The 20 OTUs with the highest abundance in the RB07 bore made up 25.8% of the total abundance in the bore, 16 of the top 20 were Proteobacteria (23.4%) (Figure 2). Within the Proteobacteria phylum were 11 OTUs of Fe oxidising bacteria from the *Gallionellaceae* family (18.5%), three OTUs of *Burkholderiaceae* and two OTUs of S oxidising, $NO_2^-$ reducing bacteria (Genus *Sulfurifustis*) [48]. Additionally, there were two OTUs from the Nitrospirota phylum capable of $NO_2^-$ oxidation and $SO_4^{2-}$ reduction (Order *Thermodesulfovibrionia*) [49], an OTU from the Chloroflexota phylum capable of organohalide-respiration (Order *Dehalococcoidia*) [50,51] and an OTU from the Actinobacteria phylum that has no relevant functions (Class *Solirubrobacterales*) [52]. Bore RB07 is paired with RB06; Fe oxidising bacteria dominated the most abundant microorganisms in bore RB07 with some S and N related OTUs.

3.4.2. Ammonia and Nitrogen Contaminated Bores (ANCB)

*Bore RB12*

Most (56.1%) of the abundance in Bore RB12 was made up of four OTUs in the *Candidatus Brocadia* genus (Figure 3). The genus *Candidatus Brocadia* in the Planctomycetota phylum is one of only a few known genera capable of performing ANAMMOX [53,54]. RB12 was situated immediately downstream from the WWTP and resided within the $NH_4^+$ plume extending out of the WWTP. Bore RB12 has the highest concentrations of $NH_4^+$ of all the bores and the highest abundance of the ANAMMOX performing *Candidatus Brocadia* genus.

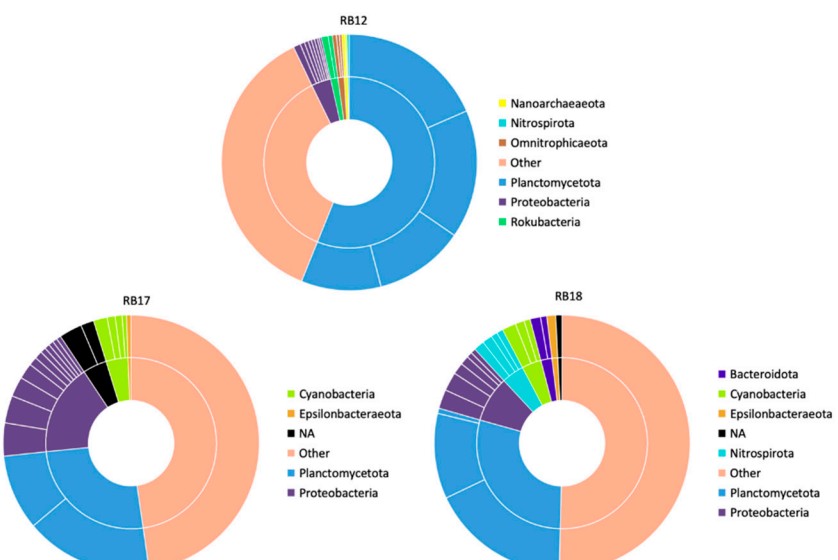

**Figure 3.** Sunburst chart of the top 50% or top 20 microorganisms (OTUs) with the highest abundance in the ammonia and nitrate contaminated bores (ANCBs) (RB12, RB17, RB18) separated by bore and organised by phylum. 'Other' represents the proportion of OTUs not represented by the top 50% of the most abundant OTUs. The inner ring of the sunburst chart represents the phylum taxonomic level and the inner ring represents the individual OTUs.

*Bore RB17*

A significant fraction (25.6%) of the abundance in bore RB17 was made up of two OTUs in the *Candidatus Brocadia* genus (Figure 3). In terms of abundance,16.7 % was made up of 10 OTUs in the Proteobacteria phylum; two OTUs of *Rhodoplanes*, two OTUs of *Pseudomonas*, an OTU of *Stenotrophomonas*, two OTUs of *Enterobacteriaceae* which is a pathogenic $NO_3^-$ reducer, two OTUs of *Immundisolibacter* which is capable of aromatic compound degradation [55] and an OTU of *Sphingomonadaceae*. In addition, 3.6 % of the abundance was made up of three OTUs of Cyanobacteria, an OTU of *Melainabacteria* and two OTUs of *Sericytochromatia* which is capable of N fixation [56,57]. Finally, two unidentified microorganisms accounted for 4.6% of the abundance. Bore RB17 was located within the Brown's Road Farm downstream from the WWTP. Like bore RB12, a large proportion of the abundance in RB17 is made up of ANAMMOX OTUs and much of the rest of the OTUs are comprised of pathogenic bacteria and OTUs related to the N cycle.

*Bore RB18*

The 20 OTUs with the highest abundance in the RB10 bore made up 49.6% of the total abundance in the bore (Figure 3). The Proteobacteria phylum (8.6%) was represented by 6 OTUs; two OTUs of *Pseudomonas*, two OTUs of *Enterobacteriaceae*, an OTU of *Immundisolibacter* and an OTU of Sphingomonadaceae. The Planctomycetota phylum (29.0%) was represented by two OTUs of *Candidatus Brocadia* (28.4%) and an OTU of the *Phycisphaeraceae* family known to reduce N [58]. Additionally, there were four OTUs of *Nitrospira*, two OTUs of *Sericytochromatia* and an OTU of *Melainabacteria*. Finally, the Epsilonbacteraeota phylum had an $NO_3$ reducing OTU (genus *Arcobacter*) [59], and the Bacteroidota phylum had two environmental OTUs (genus *Citreitalea*). Bore RB18 was paired with RB17 and like bores RB12 and RB17, ANAMMOX OTUs made up a large proportion of the abundance in bore RB18 and much of the rest of the OTUs comprised of pathogenic bacteria and OTUs related to the N cycle.

### 3.4.3. Uncontaminated Bores (UB)

*Bore BS-002*

A high proportion (45.3%) of the abundance in BS-002 was made up of 14 OTUs of Proteobacteria (Figure 4). Within the Proteobacteria phylum there were two OTUs of denitrifiers (Genus *Azospirillum*) (14.6%) [60], two OTUs of Fe, $SO_4^{2-}$ and $NO_3^-$ reducers (Genus *Magnetospirillum*) (7.6%) [61], two OTUs of S oxidising, $NO_3^-$ reducing bacteria (Genus *Sulfuritalea*) (8.2%) [62], two OTUs of *Burkholderiaceae* (6.5%), two OTUs of S reducers (Genus *Desulfatirhabdium*) [63], two OTUs of Fe and $NO_3^-$ reducing bacteria (Genus *Ferribacterium*) [64] and two OTUs of methanotrophs (Genus *Methylomonas*) [65]. Additionally, the Euryarchaeota phylum had two OTUs of methanotrophic bacteria (Genus *Methanospirillum*) [66] and the Epsilonbacteraeota phylum had two OTUs of S oxidising $NO_3$ reducing bacteria (Genus *Sulfuricurvum*) [67]. The BS-002 bore was located within the Tootgarook Swamp, with no known groundwater contamination influence from the WWTP or agriculture. Most of the abundance in the BS-002 bore was focused on denitrification reactions with some S and Fe redox reactions.

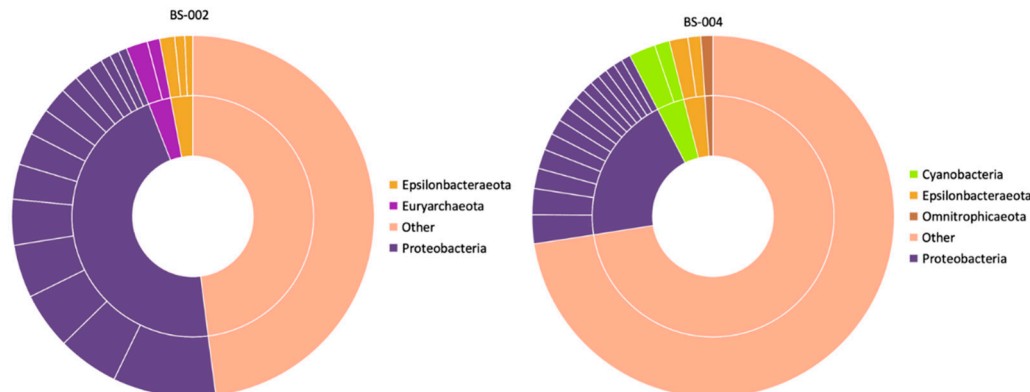

**Figure 4.** Sunburst chart of top 50% or 20 microorganisms (OTUs) with the highest abundance in the uncontaminated bores (UBs) (BS-002, BS-004) separated by bore and organised by phylum. 'Other' represents the proportion of OTUs not represented by the 20 most abundant OTUs. The inner ring of the sunburst chart represents the phylum taxonomic level and the inner ring represents the individual OTUs.

*Bore BS-004*

The 20 OTUs with the highest abundance in the RB07 bore made up 27.4% of the total abundance in the bore, 16 of the top 20 were Proteobacteria (19.8%) (Figure 4). Within the Proteobacteria phylum were three OTUs of *Sphingomonadaceae*, two OTUs of *Enterobacteriaceae*, two OTUs of *Immundisolibacter*, two OTUs of *Burkholderiaceae*, two OTUs of *Desulfobulbaceae*, an OTU of *Pseudomonas*, an S oxidising, $NO_3^-$ reducing bacteria (Genus *Sulfuricella*) [68], a methanotrophic $NO_3$ reducer (Genus *Methylobacillus*) [69] and a $SO_4^{2-}$ reducing bacteria (Genus *Desulfatitalea*) [70]. Additionally, there were two OTUs of *Sericytochromatia*, two OTUs of *Arcobacter* and one $NO_3^-$ reducer from the Omnitrophicaeota phylum (Genus *Omnitrophaceae*) [71]. The BS-004 bore was also located within the Tootgarook Swamp slightly further downstream than BS-002. The most abundant microorganisms in BS-004 were focused on N and S processes, with a few opportunistic pathogens and other processes scattered throughout.

## 4. Discussion

Previous sampling data from this site and the physicochemical data obtained indicate that the UBs are the most likely bores to reflect the background for the study area, with no notable $NH_3$ or $NO_3$ concentrations detected in the bores. Since their construction in 2017, $NO_3^-$ and $NH_4^+$ in the UBs remained below 0.3 mg $L^{-1}$ and 0.2 mg $L^{-1}$, respectively.

However, pairwise statistical analysis shows that microbially, there was no significant difference between the NCBs and the UBs, so either there is no difference between the microbial communities of the NCBs and the UBs or multivariate statistics were not sufficient to establish the difference in microbial communities between the two zones. This dataset provides a rare opportunity to observe different contamination conditions in a single aquifer and understand the microbial differences between uncontaminated bores and different types of contamination (WWTP and agricultural pollution) spread across other bores within a range of physicochemical conditions.

Table 3 shows the richness, diversity, and evenness estimators and the percentage commonality of OTUs between bores. Despite all this information and the multivariate statistical analysis none of this data analysis shows any significant trends that enable the differentiation between the uncontaminated and the contaminated bores. This leads to the conclusion that statistical analysis by itself, with the currently available tools, may not be an effective way to differentiate between the microbial communities in uncontaminated and contaminated conditions. A more appropriate analysis of background communities may involve a combination of indicator species and a deeper understanding of the major ecosystem functions being performed in the background communities compared to contaminated communities.

### 4.1. Major Ecosystem Function Analysis

Analysing the major ecosystem functions allows us to delve into trends that may have been previously unnoticed, enabling us to build profiles of the ecosystems in different areas and different levels of contamination and non-contamination. We can then review these profiles against each other and determine the similarities and differences. With this method (possibly with some standardisation), we can build a network of profiles that enables comparison of similarities and differences across areas and regions. Trends such as major ecosystem functions that consist of many species, a large percentage of the abundance, or a combination of the two could be used to indicate a disturbance in the ecosystem. Indicator species and groups are useful in this regard; however, the ecological function may not be tied to a specific species or group but may have interchangeable species dependent on the physicochemical properties and location.

In our previous work [16], an $NH_4^+$ plume extending from the WWTP was detailed by analysing the physicochemical, diversity, evenness, and phylum data. As a result of the microbial community analysis, the presence of undetected $NH_4^+$ contamination in RB17 was hypothesised because it showed similar characteristics to the other $NH_4^+$ contaminated bores. Major ecosystem function (MEF) analysis builds upon this, demonstrating the similarities between the $NH_4^+$-contaminated wells. The analysis also shows some of these characteristics in the RB06 bore, namely the presence of *Candidatus Brocadia*, though with much lower abundance. We hypothesise that this may be an early indication that the $NH_4^+$ contamination extending from the WWTP has begun to influence the microbial communities in the RB06 bore. *Candidatus Brocadia* was also present in even lower abundance in RB10, which may indicate a similar process. There were no indications of any $NH_4^+$ contamination events in the physicochemical legacy data for RB06, RB10 or RB17 in the past 10 years and all bores are adjacent to the known $NH_4^+$ contamination plume. This shows the usefulness of using this type of analysis in addition to that of the physicochemical analysis.

The DSE63273 bore is also a deep (25.1 m) bore with high concentrations of $NO_3^-$; thus, the high abundance of N-related OTUs was expected. The chemistry data in Table 1 show no indication of other contamination; however, DSE63273 is adjacent to a road and immediately downstream from a manure stockpile. This MEF analysis shows a large number and abundance of methanotrophs and S-reducing bacteria present in the bore. Roads such as the one immediately adjacent to DSE63273 are known to impact the local ecology in their surrounds. Previous studies have shown that methanotrophs are associated with hydrocarbon contamination, and asphalt roads are known to contain high concentrations of S [72,73]. Sulphur reducers and methanotrophs could also originate from the manure stockpile. It seems more likely that

the sulphur reducers and methanotrophs originate from the manure stockpile; however, the influence of the road cannot be ruled out completely.

The MEF analysis also shows many Fe-reducing bacteria in the RB07 bore; this was reflected in the physicochemical data, with RB07 having one of the highest concentrations of Fe across all the samples. In this example, the high abundance of Fe-reducing bacteria in the bore could be a result of the high Fe concentration combined with the slightly reducing environment, creating a more favourable environment for Fe reduction. BS-004 also has high Fe; however, it does not show a comparable abundance of Fe-reducing bacteria. This may be due to the differences in redox potential, EC and pH between the bores.

There are two bores in this project that were identified as possible background bores. Locationally, both bores were considered to be outside the influence of the WWTP contamination, and the physicochemical analysis (Table 1) shows low concentrations of both $NO_3^-$ and $NH_4^+$. Table 3 shows very few common OTUs between the two bores, though the difference in depth and redox conditions may play a large role in this.

### 4.2. Determining Microbial Background

The first step in differentiating between impacted and unimpacted microbial communities should be to define any anthropogenic activities or contaminants in the region and determine if these may have impacted the ecosystem. For this study, the two main contaminants of concern were $NO_3^-$ and $NH_4^+$; we propose that if a large percentage of the MEF was attributed to N transformation in the uncontaminated bores, this might be cause for concern. The MEF analysis showed that of the two UBs (within the top 20 OTUs or 50%), bore BS-002 attributed 41.7% of its abundance to microorganisms capable of N reduction and bore BS-004 attributes 14.5% of its abundance to microorganisms capable of N transformation. The bores immediately downstream from the WWTP also showed the presence of several pathogenic OTUs, which can also be seen in BS-004. The MEF analysis for bore BS-004 showed a diverse range of OTUs and functions; there does not seem to be any influence of $NO_3^-$ on the assemblage of the microorganisms observed in the well, and both the Pseudomonas and Enterobacteriaceae found in the bore had low abundance and naturally occur in groundwater [74]. Due to this evidence, we propose that BS-004 may be a suitable representative of an unimpacted microbial community for the area observed in this study. Bore BS-002 also showed a diverse range of OTUs with a range of functions; many of these OTUs perform N transforming functions which could indicate intermittent or past N contamination in the bore. There are few legacy data for the BS-002 bore, so no conclusions about past contamination can be gleaned from this; however, the bore was adjacent to an equestrian centre and at the edge of the Tootgarook Swamp closest to the market gardens (agricultural area) so contamination events cannot be completely ruled out.

Hypothetically, if the composition of BS-002 was a result of N contamination in the region that would mean that the microbial community could adapt and remove that N at a rate that left it undetected through physicochemical testing. If this were the case, the value of such an ecosystem service would be immeasurable. However, further analysis of the microbial communities would be needed to have any conclusive results regarding this.

### 4.3. Concluding Remarks

Towards the first aim, we have shown evidence of $NH_4^+$ contamination corresponding with an increase in the percentage of abundance occupied by the Planctomycetota phylum (specifically the *Candidatus Brocadia* genus). The trends across the $NO_3^-$ contamination were less clear, with no specific organism/s showing repeated increased abundance across the NCBs. In general, the NCBs seemed to have an increase in the abundance of denitrifying and N-reducing OTUs; however, more analysis of the microbial communities is needed for a more conclusive outcome.

For the second aim, using a physicochemical and microbial community dataset taken over a relatively short sampling campaign, we have demonstrated a method to differentiate between bores impacted by $NH_4^+$ and those unimpacted by $NH_4^+$. Out of the bores we

sampled, the bore most likely to represent an unimpacted microbial ecosystem was shown to be BS-004 whereas bore BS-002 showed signs that it might have been impacted by nitrogen contamination in some capacity.

For the final aim, we provided evidence that at the sampled site bores previously thought to be outside the influence of $NH_4^+$ contamination may be experiencing the influence of $NH_4^+$ contamination. This shows that there is value in using functional group analysis alongside routine physicochemical analysis to provide greater breadth and depth of understanding of the extent and impact of contamination in an aquifer.

Future research may be best aimed at improving our understanding of the interactions between contamination and microbial ecosystems in groundwater; this could be achieved by further expansion of this work by including of more sites and longer, more frequent sampling campaigns. Future research could also focus on appropriate statistical methods with the power and breadth to differentiate between different ecosystems and could include the use of identification tools such as PICRUSt2 or Vikodak.

**Supplementary Materials:** The following supporting information can be downloaded at: https://www.mdpi.com/article/10.3390/environments9100128/s1, Table S1: Methods of analysis used by ALS; Table S2: R studio packages used throughout analysis.

**Author Contributions:** Conceptualization, methodology, software, formal analysis, investigation, data curation and writing—original draft preparation was all completed by J.G.M. Supervision and project administration were the responsibilities of A.S.B., M.J.C., S.M.R., A.S. and M.M. Funding acquisition: A.S.B., M.J.C., S.M.R., A.S., M.M., N.D.C., S.A. and W.R. Writing—review and editing was completed by A.S.B., M.J.C., S.M.R., A.S., M.M., N.D.C., S.A., D.H., W.R. and J.G.M. Comments and edits were organised and enacted by J.G.M. All authors have read and agreed to the published version of the manuscript.

**Funding:** This research was funded by CRC CARE with in-kind support from Gippsland Water, Melbourne Water, South East Water, Western Water (Grant No. 2.3.02).

**Data Availability Statement:** 16S microbial community data and environmental data can be found in the NCBI Sequence Read Archive (SRA) (https://www.ncbi.nlm.nih.gov/bioproject/PRJNA861729, accessed on 13 September 2019) under the heading 'Groundwater microbial communities in an unconfined nitrogen contaminated aquifer'.

**Acknowledgments:** The authors would like to thank CRC CARE, Gippsland Water, Melbourne Water, South East Water, Western Water and BlueSphere Environmental for their funding and support in undertaking this work (Grant no. 2.3.02). The authors would also like to thank Esmaeil Shahsavari for his generous assistance with the sequencing and Kirill Tsyganov and the Monash Bioinformatics Platform for their assistance with the analysis of the microbial community data.

**Conflicts of Interest:** The authors declare that they have no known competing financial interests or personal relationships that could have appeared to influence the work reported in this paper.

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
