# Peer review of "Differentiation between Impacted and Unimpacted Microbial Communities of a Nitrogen Contaminated Aquifer"

_environments, doi:10.3390/environments9100128_

Round 1

Reviewer 1 Report

General comment

Although I agree with the authors that microbial function would be a better measure to determine the background microbial community, the approach they used may be heavily biased. This would require complete re-analysis using analytical tools like PICRUSt2 or Vikodak.

The introduction including the aim statements is nice for the discussion of background microbial communities but somewhat irrelevant to what this study is about. The use of PERMANOVA seems to be flawed as well. I also noticed that there are substantial “discussions” in the Result section (e.g., L290-311). Please reorganize them into the Discussion section.

Please see the specific comments below and modify them.

Specific comments

Abstract: Please add a little bit of background to introduce the study.

L30 and 34: Just specify the names of analyses used. Both MANOVA and PERMANOVA contain “multivariate” in their names.

L32: Add “the physicochemical data (or status) of” before “three areas of contamination”

L90-91: I agree with the authors about the reasons for the lack of microbial background in ecological studies. Perhaps I can add a related issue of the transient nature of microbial communities. However, because of the dynamic and complex nature of microbial communities, it is not easy, if not impossible, to identify the microbial background. For example, studies have shown (I’m reviewing this at home, so can’t find a specific reference) that environmental restoration often won’t result in the restoration of microbial communities, rather they usually “evolve” into the different status of microbial communities. The idea is relevant by itself, but it doesn’t seem to be relevant to the study as the listed aims also don’t address this problem. The mere comparison between impacted and unimpacted microbial communities is not novel at all. I would tone down this section to make it more relevant to the current study.

L100: Consider calling “6,000 microbial populations”, instead of “6000 microorganisms”.

L130-133: This is a very good aim statement because I think function could be another useful measure than a structure for the study of background microbial communities, however I don’t think the current study addressed it in a very meaningful way. On the other hand, the microbial background would be useful due to its sensitivity to the physicochemical background, then I’m having a hard time seeing the value of microbial functions derived from the known physicochemical background.

L136-137: Properly cite and save space here.

L194-195: Remove this statement and move table 1 to the Result section.

L228-229: Why did authors put s after MANOVA and PERMANOVA? You can simply say “MANOVA was used to…”

L232-234: Specify the procedure of OTU table preparation in section 2.5.3. to confirm this statement. Or just remove this statement, since the reason that we use an operational taxonomic unit is to work around the difficult issue of the microbial species concept.

L234-236: How about then ASV? With the reason following right after, I agree that using OTU is acceptable for purpose of the current study, so remove this statement.

L255-258: What is the context of this data? Are they before the nearby agricultural establishment and WWTP construction?

L258-261: Explain the different patterns of distribution for ammonia nitrate and total N.

L270 & 280. Just spell out MANOVA and PERMANOVA. Perhaps it would be better to do that in the Method section.

L282 and Table 2: I see the authors reported f value from PERMANOVA. This is a “pseudo F value” and just like the F value of ANOVA (and MANOVA), it indicates the effect size differences so a very low f (like 0.0002) can’t indicate a significant difference. I suspect you were confused with the p value. Consult your in-house statistician and make the appropriate change.

L286-287 and Table 2: Explain this analysis with ammonia. Table 2 indicates this is from PERMANOVA but I can’t understand the meaning of groups under Ammonia.

L296-297: An ordination plot would be a nice way to show this. I would try NMDS since it is based on distance and similar in idea to ANOSIM, which is similar to multivariate analysis like PERMANOVA.

L290-311: This section may better serve in the Discussion section.

L305-308: I can’t agree with this statement. As discussed above, it is perhaps more due to the complex and transient nature of microbial communities especially their compositional dimension not necessarily due to the existing analytical approaches, especially since the authors haven’t exhausted possible approaches and incorrectly used some they used here. But I agree with the conclusion of the section (L308-311) for the reason above. So, modify this section appropriately.

Figure 2. Add what those two circles of each chart represent. I assume they represent hierarchical taxonomy, but figure legend should contain all needed info to read the figure.

Section 3.5. I agree with the need to investigate ecosystem functions, however I’m not very sure if the approach the authors’ took was the most informative. First, the top 20 most abundant OTUs represent less than half of the OTUs identified except ANCBs. Second, the listed known functions of the identified OTUs are by no means exhaustive. Thus I feel that the findings seem to be biased per the authors’ intention of contamination zones. Still limited by their nature, I would accept with more confidence the results of simulated functional potential from 16S amplicon sequence data such as PICRUSt2 or Vikodak.

L404-405: I think there are a few more available now such as Kuenenia, Jettenia, and Scalindua.

L476-479: Absolutely! However, I’m not so sure the manual and selective approach used in the current study enables this.

L482-495 and others in 4.1.: This deterministic idea can’t justify the usefulness of the microbial background over the physicochemical background.

4.2.: The qualitative argument is fair, but some kind of measure or parameter to indicate the suitability of microbial background would be nice too. Since other bores are heavily influenced by N contamination, diversity and evenness indices may be a good candidate to try.

Reviewer 2 Report

The ms "Differentiation between impacted and unimpacted microbial cvommunities of a nitrogen contaminated aquifer" by Morrissy and collaborators is an interesting paper with interesting results, but the authors should improve its presentation.

- Line 45, is not clear what characterization means

-Line 62, it will be convenient to mention the period of time between both values

- Lines 116-133, authors should condensate the aim of the ms which has been mentioned all along the introduction.

- Line 164, it will be convenient to know why some samples couldn´t be obtained

- Line 167 and all along the text, authors of the refered paper shouldn´t be mentioned. Morrissy and colaborators should be more than enough because all the authors are mentioned in the corresponding reference.

- Table 1, in my copy is difficult to read the table, appears in three pages. The interval values for the only sample does not make any sense. Table should be improved. Metall elements could be shown as Supplemenatry Information.

- Line 290, which site?

- Table 3, intervals for only one sample doesn´t make any sense

- Figure 2, authors should consider the new aproved names for phyla

- Lines 329 to 394, it will be very convenient to have a Table showing the main differences between bores.

- In my copy, and is a pdf, I have overlaping of text and figures. 

- Lines 402 to 468, as above it will be convenient to hhave a table showing the main differences between the mentioned bores.

- Line 510, BS04 has a higher concentration of Fe and it is not clear if rge mentioned reduction of Fe applies.

Reviewer 3 Report

The author(s) utilized a dataset that provides a rare opportunity to observe different contamination conditions in a single aquifer and understand the differences between potential background bores and two different types of contamination spread across the other bores. The results are interesting and in line with the study. The authors have excellent knowledge of the topic. However, some minor revisions are necessary before accepting the article for publication.

(1)   Overall, the results are only presented and not well compared with the findings of other researchers. I think the discussion section can be improved.

(2)   There are some typo errors in the manuscript, the authors should check carefully and correct all the errors to meet the journal standard.

Round 2

Reviewer 1 Report

See a direct copy of the author's response. I want authors to spend some time and make sure what they said in the author's response corresponds to the main text.

L404-405: I think there are a few more available now such as KueneniaJettenia, and Scalindua.

Response: We agree with the reviewer view and have therefore added the additional information as requested.

 They said they made change but the change was never made to the main text.
